# Dual Geometry Schemes in Tetrel Bonds: Complexes between TF_4_ (T = Si, Ge, Sn) and Pyridine Derivatives

**DOI:** 10.3390/molecules24020376

**Published:** 2019-01-21

**Authors:** Wiktor Zierkiewicz, Mariusz Michalczyk, Rafał Wysokiński, Steve Scheiner

**Affiliations:** 1Faculty of Chemistry, Wrocław University of Science and Technology, Wybrzeże Wyspiańskiego 27, 50-370 Wrocław, Poland; mariusz.michalczyk@pwr.edu.pl (M.M.); rafal.wysokinski@pwr.edu.pl (R.W.); 2Department of Chemistry and Biochemistry, Utah State University, Logan, UT 84322-0300, USA

**Keywords:** trigonal bipyramid, MP2, NBO, MEP, σ-hole

## Abstract

When an N-base approaches the tetrel atom of TF_4_ (T = Si, Ge, Sn) the latter molecule deforms from a tetrahedral structure in the monomer to a trigonal bipyramid. The base can situate itself at either an axial or equatorial position, leading to two different equilibrium geometries. The interaction energies are considerably larger for the equatorial structures, up around 50 kcal/mol, which also have a shorter R(T··N) separation. On the other hand, the energy needed to deform the tetrahedral monomer into the equatorial structure is much higher than the equivalent deformation energy in the axial dimer. When these two opposite trends are combined, it is the axial geometry which is somewhat more stable than the equatorial, yielding binding energies in the 8–34 kcal/mol range. There is a clear trend of increasing interaction energy as the tetrel atom grows larger: Si < Ge < Sn, a pattern which is accentuated for the binding energies.

## 1. Introduction

The continuing development of modern chemistry is leading to improved understanding of microscale processes. An important part of this knowledge accrues to the rapidly accelerating understanding of noncovalent interactions. necessary to describe chemical and biological events occurring in living organisms as well as in vitro environment [1,2,3]. While the H-bond is certainly the most common and best understood noncovalent bond, it is becoming increasingly recognized that there are very similar interactions in which the bridging H is replaced by any of a wide range of other atoms. For example, the very electronegative halogen atoms may play the role of bridge in the eponymous bonds [4,5,6,7,8,9]. A central tenet of halogen and related bonds is the σ-hole [10,11,12,13,14,15] which refers to an electron-depleted region that lies directly opposite a covalent bond. With its positive electrostatic nature, this hole can attract a nucleophile [14,16,17], and the entire interaction is supplemented by polarization and dispersion forces. This idea is not limited to halogens, but has been extended to other electronegative atoms [18,19,20] and has recently expanded to transition metal interactions with Lewis bases, sometimes referred to as a “regium bond” [21]. It is not only the plane of the molecule which can house a positive region. When this phenomenon occurs above the plane of a molecule, it is commonly dubbed a π-hole [22,23,24].

The tetrel family of atoms (C, Si, Ge, etc.) is amongst those that can act as bridges [25,26,27,28] in analogies to H-bonds. The importance of tetrel bonds has been explored in a variety of forums including the S_N_2 reaction [29], stabilization of crystal structures [30,31], ability to serve as anion receptors [32], complexation of carbon dioxide with azoles [33] or cyclopenta- and cyclohexatetrelanes with selected anions [34]. In the biological realm [35] this sort of bond appears to be common in proteins [36,37] and involved in the enzymatic activity of S-adenosylmethionine-dependent methyltransferases [38]. Tetrel bonds were also considered in terms of their cooperativity [39], characterization in the solid-state [40], and interactions in general biological systems [41].

Halogen, chalcogen, and pnicogen atoms are typically covalently bonded to one, two, and three substituents, respectively. They thus offer little in the way of steric crowding to hinder the approach of a nucleophile. However, calculations have shown [42] how the larger number of substituents that accompany hypervalency of any of these sorts of central atoms can offer an impediment to noncovalent bonding and complicate the analysis of contributing factors. Even in its normal bonding situation, the tetrel atom is surrounded by four substituents. In such a case, the Lewis acid must significantly deform in order to provide a space for the nucleophile, accompanied by large deformation energies. This rearrangement has been shown to dramatically affect the molecule’s electrostatic potential, with ensuing repercussions for the interaction with a base.

These considerations lead to a central question. Can the internal rearrangements within a tetrel-containing molecule lead to alternative geometries for the complex with the base? While it is understood that a σ-hole is common along the extension of a R–T bond (where T and R represent a tetrel atom and a neighboring substituent atom) what would happen as the normal tetrahedral arrangement of the TR_4_ molecule is deformed so as to accommodate the base? A planar D_4h_ sort of arrangement would preclude any σ-holes, leaving only the possibility of a π-hole. However, such a planar bonding situation is unlikely given its very high energy. If the TR_4_ were to remodel itself toward a trigonal bipyramid, as seems more likely, the vacant fifth site, awaiting the nucleophile, could lie at either an axial or equatorial position. The former represents the most common situation observed to date, wherein the base lies directly opposite one T–R bond, along a σ-hole, with the other three bonds occupying what might be termed equatorial sites. But one could certainly envision an alternative scenario, where the base occupies a vacant equatorial position. It is the goal of this work to consider these two options, to identify which is the more stable, and to understand the reasons for this preference. Another objective is the ability to predict in advance what particular sorts of systems will tend toward one geometry over the other. What conditions are necessary for both of these structures to coexist, and which pairs of molecules will only be present as one or the other?

In order to examine a range of different tetrel atoms, perfluorinated SiF_4_, GeF_4_, and SnF_4_ were taken as the Lewis acids. Pyridine and several of its *p*-substituted derivatives were used as Lewis base. Methyl and OH substituents were used to assess effects of electron-donating groups, and electron-withdrawing CF_3_ to analyze the opposite influence.

## 2. Results

### 2.1. Electrostatic Potentials of Monomers

The molecular electrostatic potential (MEP) of the various TF_4_ monomers all have the same characteristics, as displayed in the top part of Figure 1. Within the context of a tetrahedral geometry there is a negative region surrounding each of the F atoms, and a positive area that lies directly opposite each T–F bond, known generally as a σ-hole. The magnitude of this hole can be characterized by the value of the maximum on a surface of isodensity, V_s,max_, in this case ρ = 0.001 au. The values of V_s,max_ listed in Table 1 rise as the tetrel atom enlarges from a minimum of 41.3 kcal/mol for T = Si to a maximum of 70.1 kcal/mol for Sn. The MEPs of the four Lewis base derivatives of pyridine are also presented in Figure 1, and all contain a particularly negative region that is coincident with the N lone pair direction. As may be noted in the lower half of Table 1, the minimum on the 0.001 a.u. isodensity surface lies in the range between −30 and −38 kcal/mol; it is smallest for the R = CF_3_ substituent, which can be connected with the electron-withdrawing capacity of this species. 

### 2.2. Structures of Complexes

The approach of the Lewis base to the originally tetrahedral TF_4_ can be thought of as the addition of a fifth ligand. The complex thus takes on some of the geometric characteristics of the penta-coordinated trigonal bipyramid. As such, the base can occupy either an axial or equatorial position on this framework. For most of the systems considered here, both such positions represent a minimum on the potential energy surface. These two geometries are displayed in Figure 2 and are denoted according to the axial or equatorial position of the base. The geometric aspects of the optimized complexes are reported in Table 2. As expected, the N atom lies very nearly directly opposite the F_ax_ atom in the axial complexes, with θ (N··TF_ax_) close to 180°. The angle between the N and the three equatorial F atoms is a bit less than the idealized 90° in a perfect trigonal bipyramid, due to the longer R (T··N) distance, as compared to R (T–F_ax_). The latter angle is of course quite different than the 109.5° in the uncomplexed tetrahedral TF_4_, suggesting a substantial monomer deformation upon complexation. With regard to the equatorial complexes, the N atom lies very close to 120° from the two equatorial F atoms. The two axial F atoms are both bent in a bit towards the N atom, leaving the θ (N··TF_ax_) angle less than the idealized 90° by some 5–7°.

The axial complex is preferred over the equatorial structure in every case, and the margin of this energetic difference is contained in the last column of Table 2. This difference varies between 2.7 and 9.4 kcal/mol. It is greatest for Si and smallest for Sn, but shows little dependence upon the pyridine substituent R. It might finally be noted that in the single case of the 4-TFMPy∙∙∙SiF_4_ complex, with T = Si and R = CF_3_, there is no minimum corresponding to the equatorial geometry.

One may glean some insights into the energetics of the final complexes by consideration of the characteristics of the TF_4_ monomer, when organized in the trigonal bipyramid structure to which it tends within the complex. That is, each monomer was optimized but with the θ (F–T–F) angles characteristic of the trigonal bipyramid (90°, 120°, 180°) but with a vacancy in either an axial or equatorial position. There is a marked preference for an axial vacancy, as indicated by the second column of Table 3. This preference is equal to 10.7 kcal/mol for SnF_4_ and doubles as the tetrel atom is reduced in size to Si. Just on the basis of the energetics of the monomer, then, one would anticipate that the axial complexes with the pyridine derivatives should be more stable than the equatorial structures.

Each of the two monomer structures will have its own set of σ-holes, i.e., maxima in its MEP. The values of V_s,max_ in the vacant site listed in the last two columns of Table 3 are slightly larger for the equatorial vacancy. Another important issue compares the intensities of the σ-holes in the idealized trigonal bipyramid structures with those in the tetrahedral geometries of the unperturbed monomers. The values of V_s,max_ for the penta-coordinated structures in Table 3, both axial and equatorial vacancies, are far larger than those in the tetrahedral molecules in Table 1. This intensification of the σ-hole, and the ensuing magnification of the electrostatic attraction with an incoming nucleophile, provides an important impetus for the geometrical distortion of the TF_4_ that accompanies complexation.

An important facet of the geometries resides in the intermolecular distance R (T··N). These distances are contained in Table 2 and are in the range between 2.0 and 2.3 Å. In the axial structures, the distance is longest for T = Sn, understandable in light of its larger size, but it is also slightly longer for Si than for Ge, which can be understood in terms of energetics (see below). With respect to pyridine substituent R, the intermolecular distance elongates in the order CF_3_ > H > Me ~ OH, consistent with the pattern of more negative V_s,min_ in Table 1. 

R (T··N) is a bit shorter for the equatorial structures as compared to axial; this decrement is sensitive to the nature of the tetrel atom. The closer contacts within the equatorial complexes is an important characteristic that has implications on the energetics discussed below. In short, an equatorial site within a trigonal bipyramid suffers from lesser steric repulsions than does an axial position. The Lewis base is thus freer to approach the tetrel atom from the equatorial direction, resulting in the smaller values of R in Table 2. Note that the difference in R between equatorial and axial dimers is equal to 0.17, 0.10, and 0.03 Å for T = Si, Ge, and Sn, respectively. This diminishing difference reflects the reduced steric repulsions for the larger tetrel atoms that move the F atoms further apart. The same effect is responsible for the lower energetic preference of the axial over the equatorial dimers for the larger T atoms in the last column of Table 2. The dependence on substituent in the equatorial dimers varies in the same manner as the axial structures. There is one minor difference in that R is always smaller for Si than for Ge, as opposed to nearly equal distances for the axial structures. 

### 2.3. Energetics

The interaction energies of the various complexes reported in Table 4 show only a small dependence on the level of theory. Categories I and III both employed the cc-pVTZ basis set, (I) with MP2 and (III) with CCSD(T). The results are barely affected by the particular method of incorporating electron correlation. The switch to BLYP-D3 brand of DFT, coupled with a Def2TZVPP basis set (II) has a lowering effect on the interaction energy but only by a small amount. Most importantly, all levels of theory display the same trends from one system to the next.

The patterns in the energetic data are consistent with some of the trends in the geometries. The interactions within the axial complexes strengthen a great deal as the tetrel atom grows in size, whereas there is much less sensitivity noted in the equatorial structures, and it is the Ge that seems to be more strongly bound than either Si or Sn. The interaction, whether axial or equatorial, is strengthened slightly by either OH or Me substituent on the pyridine but weakened by CF_3_. This pattern is consistent with the idea that an electron-withdrawing agent will remove density from the N lone pair and thus weaken the interaction with a Lewis acid.

After taking proper account of the deformations that occur in each monomer as they engage one another to form the complex, one arrives at the overall energetics of the binding process, E_b_. The binding energies in Table 5 are of course less negative than the interaction energies between the previously distorted monomers in Table 4 but nonetheless show many of the same patterns. In the first place the MP2(I) and DFT(II) quantities are similar and follow the same trends, although the latter are a few kcal/mol less exothermic than the former. The intensification of the binding along the Si < Ge < Sn order is present, now in the equatorial as well as the axial complexes. Just as in the case of the interaction energies, E_b_ is also intensified for the electron-donating OH and Me substituents, but weakened for CF_3_.

But there is one very important aspect of the energetics in Table 4 and Table 5. The interaction energies of the equatorial complexes in Table 4 are considerably larger than those of the axial geometries, in some cases nearly twice the magnitude. The energetic margin of equatorial over axial is most dramatic for the Si complexes, up as high as 25 kcal/mol, and is reduced as the tetrel atom grows in size. However, quite the opposite is true of the binding energies in Table 5 where it is the axial complexes which are preferred. The margins here are a little smaller, but still reach up near 10 kcal/mol. Despite this reversal between E_int_ and E_b_, there remains the same Si > Ge > Sn ordering of the energy differences between axial and equatorial.

The opposite behavior of these two energetic measures of noncovalent bonding rests in the deformation energies. It was noted in Table 2 that the deformation of the tetrahedral TF_4_ molecule into either of the trigonal bipyramidal geometries requires quite a bit of energy. But it is more costly to leave the vacancy in the equatorial than in the axial position, in amounts varying from 10.7 kcal/mol for the larger Sn up to 22.6 kcal/mol for Si. It is the larger deformation energy of the equatorial structure that accounts for its reduced binding energy within the various complexes.

These deformation energies are recorded in Table 6, split out to show the deformation of both the Lewis acid and base individually. It is first apparent that the bulk of the deformation is associated with the Lewis acid TF_4_, since that of the base is only 1 kcal/mol or less. Secondly, the deformation required to accommodate the base in the axial position is quite a bit smaller than that for an equatorial association. The deformation energy is reduced as the tetrel atom becomes larger, which is especially true of the equatorial systems.

Comparison of Table 1 and Table 3 had indicated that in the idealized situations, the σ-hole of the trigonal bipyramid TF_4_ monomer is considerably more intense than in the optimized tetrahedral structure. The value of V_s,max_ in the geometries within the actual complexes is reported in the last column of Table 6. Since the geometries of the monomers are not quite fully formed trigonal bipyramids, the σ-holes are not quite as intense as those in Table 3, they nonetheless approach these values, and represent a huge increase over their magnitude within the tetrahedral geometry. Taking GeF_4_ as an example, V_s,max_ is equal to 50.9 kcal/mol in the tetrahedral monomer, vs. 116.4 and 120.1 kcal/mol for the idealized axial and equatorial geometries, respectively. Table 6 shows this quantity is roughly 100 kcal/mol in the axial complexes, slightly higher at 108 kcal/mol for the equatorial dimers. The increased electrostatic attraction resulting from this σ-hole magnification helps to compensate for the large geometrical deformation energy occurring within the Lewis acid molecule.

Decomposition of the total interaction energy offers a window into the factors that contribute to the stability of each complex. The various contributions are listed in Table 7 for both the axial and equatorial dimers, as well as the difference between these two structures. As an overall perspective, the electrostatic component accounts for roughly 60% of the total attractive force. The second most important factor is the orbital interaction which makes up between 34% and 41%. Dispersion makes only a minor contribution on the order of 5% or less. Focusing first on the electrostatic element, in the case of the axial dimers, it increases with heavier tetrel atom Si < Ge < Sn, while an opposite pattern is seen in the equatorial structures. The same preference for the heavier T atom occurs for the equatorial values of E_oi_, while the axial structures fall into the Ge > Sn > Si ordering. As a result, the electrostatic component favors the axial over the equatorial structure for T = Si and Ge, but tends toward the equatorial for Sn. The orbital interactions lead to a heavy preference for axial for Si, a preference which is progressively abated for Ge and then for Sn.

Much of this trending information traces back to the intermolecular distances. Each component is of course heavily dependent upon the distance between the two subunits, rising dramatically as they come closer together. As noted in Table 2, R(T···N) varies only slowly with tetrel atom for the axial complexes. Hence, the electrostatic attraction can be closely related to the value of V_s,max_ of the TF_4_ molecule, which climbs in the order Si < Ge < Sn, as does E_elec_. In the case of the equatorial dimers, R increases steadily along with the size of the T atom, thus acting to diminish E_elec_, right along with E_oi_, as T enlarges.

Previous energy decompositions of tetrel-bonded systems have observed a similar mix of terms. In one example [43], the electrostatic term was found most important for the majority of such dimers, followed by orbital interaction. It was only for the weakest complexes of CFH_3_ with π-systems (with interaction energies of only −1.3 kcal/mol) that the dispersion component played a significant role.

### 2.4. Analysis of Wave Function

One measure of the noncovalent bond strength originates in a AIM analysis of the topology of the electron density [44,45,46,47]. AIM molecular graphs for each complex are illustrated in Appendix A, where broken lines indicate intermolecular bond paths, with a small green dot marking the bond critical point. Each bond path between the two monomers is quantified via several aspects of the critical point. In particular, the density ρ, its Laplacian ∇^2^*ρ*, and the electron energy H [48], are all displayed in Table 8 for the various complexes fit into patterns that are only partially consistent with energetic and geometrical data. First focusing on the most important T···N bond path, ∇^2^*ρ* rises in the order Si < Ge < Sn for the axial complexes, the same order as observed for the interaction energies in Table 4. ρ and H display a somewhat different pattern, peaking for Ge. Consonant with the interaction energies, the AIM markers of noncovalent bond strength are larger for the equatorial complexes. Again there are different trends for ∇^2^*ρ* on one hand and ρ and H on the other. The latter are largest for Ge, while the peak of the former occurs for Si. In addition to the principal T···N bond path, there are indications of weaker bonding interactions between F atoms of TF_4_ and H of the pyridine derivatives, i.e., weak CH···F H-bonds. As measured by AIM, these bonds represent only fractions of the bond strength of T···N, but may account for a small amount of the total binding energy. 

The behavior of the AIM descriptors is characteristic of tetrel bonded and related systems. For example, quite comparable values of the density at the bond critical point were derived when NH_3_ was paired with TF_4_ (T = C, Si, Ge, Sn), as well as other less fluorosubstituted Lewis acid molecules [49]. Complexes of TFH_3_ (T =Ge, Si, Sn, Pb) with various carbon π-electron systems displayed values of ρ_BCP_ up to 0.045 a.u. and all positive Laplacians [43]. Tetrel bonds between Tr_5_Cl_10_ and Tr_6_Cl_12_ (Tr = Si and Ge) and bases HCN, HF, OH^−^ and Cl^−^ [34] noted AIM markers similar to those observed here, with similarly positive Laplacians. In particularly strong tetrel bonds with anions, one can find values of ρ_BCP_ that surpass 0.1 a.u. [50].

The delocalization indices (DI) of T···N interactions [51,52] are collected in Appendix A. These quantities specify the number of electrons in the interatomic bonding regions and can be interpreted as the bond order value [51,52]. The DI is much higher for complexes with Ge than for Si dimers which implies greater covalent nature of the former. This pattern is consistent with the other descriptors as well as interaction energies. (Values for Sn complexes were not possible to evaluate by virtue of the pseudopotential approach.)

An alternate means of analysis concerns charge transfer from the orbitals of one molecule to those of the other. Within the NBO framework, the energetic measure of such a transfer emerges as the perturbation energy E(2) [53,54,55,56,57,58,59]. The relevant transfers in these systems involve motion from the lone pair orbital of the pyridine N into the antibonding lone pair orbitals of the tetrel atom, designated LP* by NBO. (The latter can be thought of as alternate descriptions of σ* (F–T) orbitals which are centered largely on T as compared to F.) The total interorbital transfer energy is reported in Table 9. Whereas the interaction energies for the axial complexes obey a clear Si < Ge < Sn trend, E(2) is largest for Ge. The equatorial geometries lead to enlarged values of E(2) vs. axial, in common with the energetics. E(2) diminishes as the tetrel atom grows in size for these equatorial structures, inconsistent with the energetic trends.

The total charge transfers from the Lewis base molecule to the acid, encompassing all orbitals, is contained within the CT columns of Table 9. Just as for E(2), CT is much larger for equatorial than for axial complexes. With respect to the latter geometries, CT follows the Si < Ge < Sn trend of the energetics. The same can be said of the equatorial structures where CT and E_int_ both peak for Ge. Of the various NBO measures, it is the total intermolecular charge transfer that thus most closely parallels the energetics. The other general trend coming from presented analyses is that tetrel bonds studied have partly covalent nature. The amount of orbital interaction contribution presented by EDA (up to −70.50 kcal/mol), E(2) energies of intermolecular donation (reaching 169 kcal/mol), net charge transfer values exceeding 0.2 electrons and finally the negative values of H at the BCPs in AIM analysis are the apparent signs of the non-negligible influence of covalent character in interactions investigated here.

### 2.5. Conversion between Axial and Equatorial Complexes

Given the presence of two minima on the potential energy surface, with comparable energies, the issue naturally arises concerning the ease of conversion from one to the other. For this purpose, focus was placed on the unsubstituted pyridine Lewis base. The transition state was located for this transition, as illustrated in Figure 3 for the Py···GeF_4_ system as an example. The shift of the GeF_4_ unit from axial to equatorial is an interesting one. Two of the three equatorial F atoms in the axial geometry retain their position fairly steadily, with the θ (NGeF) angle remaining nearly constant at 84°. The transition occurs as the axial F atom, F_1_, in Figure 3, moves in toward the N. 

The θ (NGeF_1_) angle is 179° in the axial structure, diminishes to 132° in the transition state, until finally reaching 118° in the equatorial structure on the right side of Figure 3. In essence, it has shifted smoothly from an axial to an equatorial position relative to the pyridine. During this process, it has decreased its distance from the Ge progressively from 1.722 to 1.717 to 1.715 Å. The third equatorial F atom (F_2_) will ultimately become an equatorial ligand in the final structure. But in order to do, it must increase its angle with respect to N, with θ (NGeF_2_) beginning at 84°, increasing to 109° in the transition state, and finally to 119° in the equatorial geometry. The pertinent R (GeF_2_) bond length changes from 1.722 to 1.712 to 1.715 during this process. The barrier height G^†^ for the transition from axial to equatorial is equal to 11.79, 8.38 and 1.46 kcal mol^−1^ for Py···SiF_4_, Py···GeF_4_ and Py···SnF_4_, respectively, as reported in Appendix A. That is, the conversion becomes easier as the tetrel atom is enlarged. 

### 2.6. Other Types of Geometry

While the geometries described above represent the principal minima identified on each potential energy surface, there are additional minima albeit of higher energy. These secondary minima are displayed in Appendix A along with their salient geometric characteristics. Most of these structures involve a weak interaction between the tetrel atom and the aromatic ring, at sites other than the N atom. The aromatic atom which is directly involved in this interaction is situated directly opposite one of the T–F bonds in what might be termed an axial site. The two molecules are spread apart, with intermolecular R(T···C) distances of 3 Å or more, with Sn engaging in the strongest such interactions of those identified. In no case is there a minimum found between the TF_4_ and a C atom ortho to N, due in part to the more positive natural charge on this C relative to the others. 

## 3. Discussion

Within the context of classic ideas such as VSEPR theory, it is understood that the axial position of a trigonal bipyramid faces more steric repulsion than does its equatorial counterpart. This notion is based on the fact that each axial ligand has three close equatorial neighbors at an angle of 90°, while the equatorial site has only two such nearby neighbors. This distinction explains why the trigonal bipyramid geometry of TF_4_ with an equatorial vacancy is at an energetic disadvantage with respect to an axial vacancy. In fact, the calculated preference for the axial vacancy ranges from 10.7 to 22.6 kcal/mol (see Table 3), with the greatest difference associated with the smallest T atom Si. It is also for this reason that the deformation energies of the equatorial complexes with the pyridine derivatives are considerably larger than the same quantities for the axial structures.

The second issue where this axial vs. equatorial site distinction comes into play arises when the Lewis base approaches. For the reasons enunciated above, approach to an equatorial site will offer less steric repulsion than to an axial site with three close ligand neighbors. It is largely for this reason that the interaction energies for the equatorial complexes are more negative than for the axial geometries, and the N atom approaches more closely to the tetrel. Recall that the interaction energy involves pre-deformed monomers so it represents a pure interaction, unaffected by the energetic cost of preparing each monomer into the geometry it adopts within the dimer.

The binding energy, representing the exothermicity of the combination process, thus combines these two competing effects. It is largely the first factor, the monomer deformation energy, that controls the final energetics, making the axial complexes more stable than their equatorial counterparts. This preference varies between 3 and 10 kcal/mol, and is largest for the smallest Si atom. It is the mutual repulsions, both electrostatic and steric, that are largely responsible for the high deformation energies common to these TF_4_ molecules. For example, the replacement of some of these F atoms by H results in markedly lower deformation energies upon complexation with a Lewis base [25].

The dichotomy of two separate minima for tetrel bonded systems may be more common than previously thought. For example, replacement of the four F atoms of SiF_4_ by Br leads to a similar situation. DFT calculations show an equatorial placement of the pyridine base represents a true minimum on the surface, albeit higher in energy than its axial counterpart, by some 8 kcal/mol. 

In order to delve more deeply into these ideas, the four F atoms of SiF_4_ were replaced by bulkier methyl groups. Of course, the optimal geometry of Si(CH_3_)_4_ is tetrahedral. But when forced to adopt a trigonal bipyramid structure, the one with an axial vacancy was favored by 26.5 kcal/mol over an equatorial vacancy, due to the steric repulsions between the methyl groups. In a second step a Lewis base NH_3_ was brought up to the vacancy of each structure. When the geometry of each dimer was fully relaxed, both optimized to the same sort of structure: Si(CH_3_)_4_ became tetrahedral, with the NH_3_ fitting into a hole between the methyl groups. In other words, the instability engendered by the steric repulsions makes the monomer with an equatorial vacancy so unstable that it quickly decays to a tetrahedral geometry, a derivative of axial with a linear R–T··N. The same can be said about tri and disubstitution: When only three or two of the F atoms of SiF_4_ were replaced by the bulkier methyl group, SiMe_3_F and SiMe_2_F_2_ only engage in an axial sort of tetrel bond with a N-base, with FSi··N close to linear.

The idea that a tetracoordinated TR_4_ molecule will deform into a trigonal bipyramid has become well recognized [60,61], as well as the idea that such a deformation leads to large distortion energies [62]. The concept that a tetrel bond can form in a direction that does not coincide with a T–R bond has some precedent [63]. For example, a prior work [64] found what was termed a π-tetrel bond when XC≡N was paired with a base. However, this bond did not substantially alter the geometry of the tetrel-containing molecule. Another bonding scenario where this out-of-plane situation can occur is the planar F_2_T=O molecule [26,65,66,67] or R_2_T=CH_2_ [25]. These sorts of geometries typically result in weaker binding, as for example, the replacement of TF_4_ by TF_2_=CF_2_ (T = Si, Ge, Sn) [25]. The reduction of substituents on the T atom from four to three has another effect: it markedly reduces the steric repulsions between them, leading to much reduced deformation energy [25]. 

The interaction energies computed here are in nice accord with prior calculations of related systems. Interactions between TF_4_ molecules and NH_3_ [38] are of comparable magnitude with those computed here for the pyridine derivatives; the values are reduced slightly for pyrazine. The complexation of TF_4_ molecules with a variety of bases was studied earlier [68] but in the context of only the axial complexes. Nevertheless, the deformation of the monomers was found to strongly intensify the σ-hole which provided a strong element in the large binding energies, consistent with the trends noted here. A similar effect was also observed for the pnicogen bonds involving ZF_5_ (Z = P, As, Sn) wherein the Lewis acid transitioned from a trigonal bipyramid to a square pyramid structure.

The comparative strengths of the various tetrel atoms Si, Ge, and Sn, as well as C, has also been a subject of a good deal of prior investigation [32,49,50,69,70,71] and this sort of noncovalent bond has been found in some cases to act as a superior Lewis acid when compared to other related bonds such as chalcogen, pnicogen, and halogen types [32,70,72,73,74] although this cannot be considered a universal rule [75,76]. Deformation energies in excess of 20 kcal/mol may be a common feature of tetrel bonding. It was noted earlier [70] in complexes of TF_4_ with NH_3_, which also confirmed the rise in this quantity as the tetrel atom becomes smaller. There was also an indication that the deformation energy would be a bit larger for THF_3_. It has been shown to play a major role in the tetrel bonding of other Lewis bases, for example in those containing methyl and larger alkyl substituents [70], again with this quantity reaching up toward 20 kcal/mol. In keeping with the findings here, the deformation effects are most dramatic for smaller tetrel atoms like Si where it can prevent the formation of a tetrel bond altogether.

One last functional that was tested here was the double-hybrid B2PLYP. Calculation of the interaction energies of the three complexes involving unsubstituted pyridine displayed in Appendix A suggests that B2PLYP outperforms BLYP-D3 and yields data comparable to MP2. However, since its computational cost is comparable to actual MP2 calculations, one might be better served with MP2 itself.

Finally with regard to C as central atom, it is well known that this atom is generally reluctant to engage in tetrel bonds of any strength. Indeed, even with four highly electron-withdrawing substituents, V_s,max_ for the tetrahedral CF_4_ is only 18.8 kcal/mol, less than half that of the other TF_4_ monomers. As such, CF_4_ is unable to form equatorial complexes with pyridine or any of its derivatives. The BLYP-D3 interaction energies for the axial structures are very small, only about −1.7 kcal/mol. AIM analysis suggests that these weak complexes are held together by secondary interactions, rather than a tetrel bond.

## 4. Methods

Full geometry optimization was performed for isolated monomers and complexes at the MP2 level in conjunction with the cc-pVTZ basis set [77,78]. For purposes of confirmation, the BLYP-D3/Def2TZVPP, B2PLYP (for pyridine complexes only) and CCSD(T)/cc-pVTZ levels were employed as well [79,80,81,82,83,84]. For Sn atoms, the cc-pVTZ-PP basis set which includes relativistic effects was chosen [85,86]. All minima were confirmed by vibrational analysis, transition states were located using a standard Berny algorithm while demanding the optimization to a transition state rather than a local minimum (TS keyword). Molecular coordinates for minima of primary TF_4_ complexes with pyridine and its derivatives are given in Appendix A. The binding energy of each complex was calculated as the energy difference between the complex and the sum of the individually optimized monomers. The interaction energy takes as its reference the energies of the monomers measured in the geometries obtained within the complex. The difference between the former and latter is the deformation energy, defined as the energy required to distort each monomer from its optimized geometry to that within the dimer. The interaction and binding energies of the complexes were corrected for basis set superposition error (BSSE) via the Boys-Bernardi counterpoise procedure [87]. Geometry optimizations and energy calculations utilized the Gaussian 09 code [88]. Energy decomposition analysis (EDA) was performed at the BLYP-D3/ZORA/TZ2P level using DFT geometries with ADF software [89,90,91]. The molecular electrostatic potentials (MEPs) of the isolated monomers were calculated on the electron density isosurface of ρ = 0.001 a.u. at the MP2/cc-pVTZ level; the corresponding extrema were quantified using the WFA-SAS program [92]. MP2 electron density topology was analyzed in the context of the AIMAll program in order to visualize and characterize the bonding paths [93]. NBO analysis was employed to analyze the interactions in terms of atomic charges and orbital-orbital second-order energies using the DFT-generated wave function [94]. 

## 5. Conclusions

In summary, the approach of a Lewis base toward a tetrahedral TF_4_ molecule causes it to distort into a trigonal bipyramid shape. The base may come closer to T if it approaches along an equatorial direction, which also results in a higher interaction energy than approach toward an axial site. These interaction energies are quite high, surpassing 50 kcal/mol. One source of these large quantities is the very substantial intensification of the σ-hole that arises when the originally tetrahedral TF_4_ is distorted into a trigonal bipyramid shape. On the other hand, this same equatorial approach also induces a much larger deformation energy in the TF_4_ monomer. As a result of these two competing effects, it is the axial dimer which is the more stable of the two, by some 3–9 kcal/mol. It may be possible to isolate both sorts of structure in an experimental setting, although the energy barrier separating them is fairly low. The tetrel bond strength rises as the tetrel atom grows in size: Si < Ge < Sn; the effect of the substituent on the pyridine is considerably weaker.

## Figures and Tables

**Figure 1 molecules-24-00376-f001:**
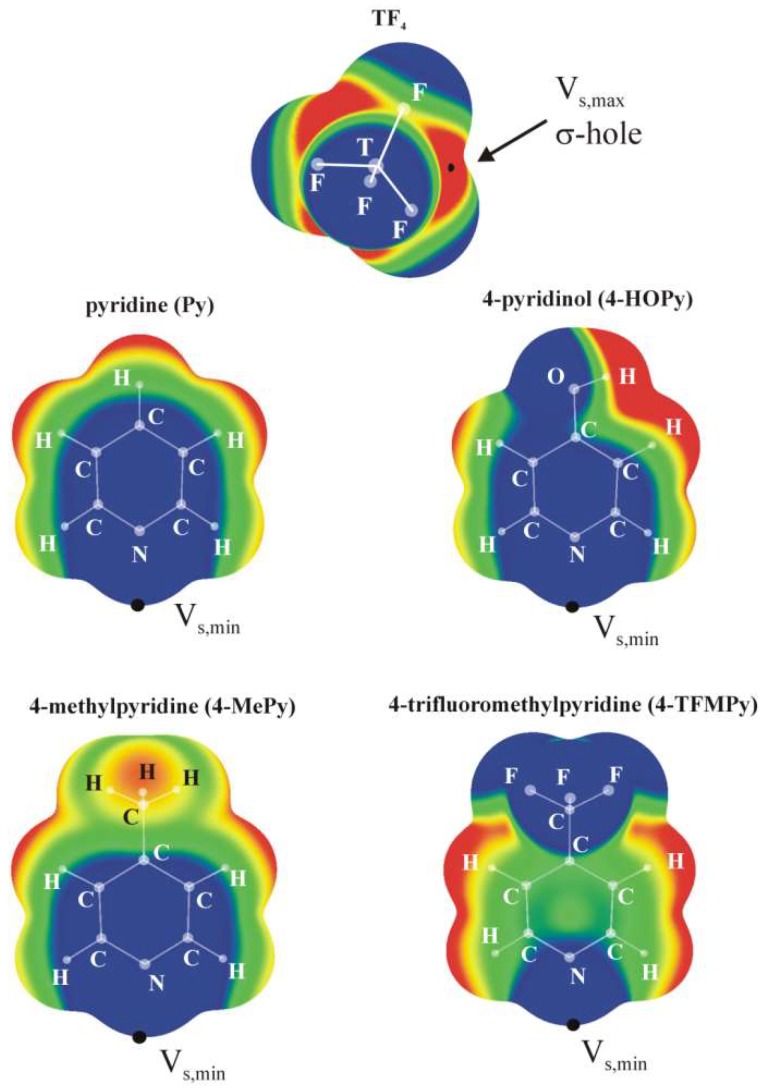
MEPs of isolated SiF_4_ and Lewis base ligands computed on the 0.001 a.u. isodensity surface at the MP2/cc-pVTZ level. Color ranges, in kcal/mol, are: red greater than 15, yellow between 8 and 15, green between 0 and 8, blue below 0 kcal/mol.

**Figure 2 molecules-24-00376-f002:**
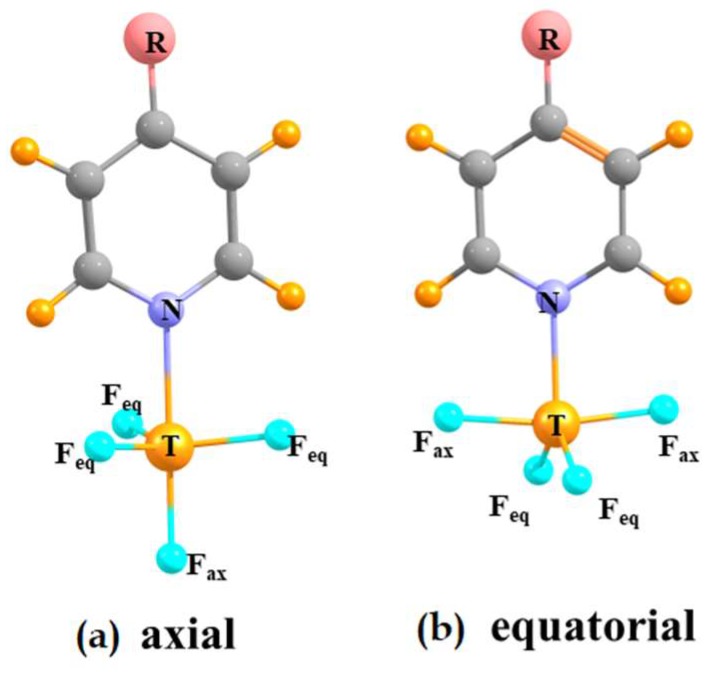
MP2-optimized structures of (**a**) axial and (**b**) equatorial complexes of TF_4_ (T = Si, Ge, Sn) with pyridine and its derivatives (R = H, OH, CH_3_, CF_3_).

**Figure 3 molecules-24-00376-f003:**
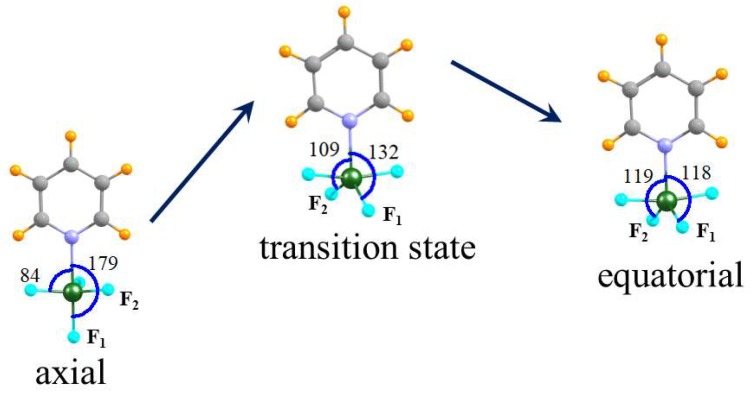
Gibbs free energy profile of transformation reaction between axial and equatorial conformers of the Py∙∙∙GeF_4_ complex. Data concerning energies and imaginary frequencies of TS are given in Appendix A.

**Table 1 molecules-24-00376-t001:** Molecular electrostatic potentials extrema (in kcal mol^−1^) on the 0.001 a.u. contour of the electron density (V_s,max_ and V_s,min_) of isolated donors of σ-hole and π-hole as well as isolated Lewis bases used in this study, calculated at the MP2/cc-pvtz level of theory.

Lewis Acid	V_s,max_ (σ-hole)
SiF_4_	+41.3
GeF_4_	+50.9
SnF_4_	+70.1
Lewis base	V_s,min_ ^a^
Py	−36.3
4-HOPy	−37.3
4-MePy	−37.8
4-TFMPy	−29.9

^a^ minimum coincides with N lone pair.

**Table 2 molecules-24-00376-t002:** Structural parameters (R in Å, angles in degs) of complexes at the MP2/cc-pVTZ level.

	Axial Complexes			Equatorial Complexes			
	R (T···N)	θ (N··TF_ax_)	θ (N··TF_eq_)	R (N∙∙∙T)	θ (N··TF_eq_)	θ (N··TF_ax_)	∆E ^a^
Py∙∙∙SiF_4_	2.143	179.6	82.9	1.973	119.1	83.7	9.42
Py∙∙∙GeF_4_	2.120	179.5	84.3	2.024	118.4	84.5	5.91
Py∙∙∙SnF_4_	2.244	178.2	83.1	2.218	118.6	83.3	3.27
4-HOPy∙∙∙SiF_4_	2.121	179.9	83.2	1.960	118.9	84.3	8.59
4-HOPy∙∙∙GeF_4_	2.104	179.9	84.4	2.012	118.4	85.2	5.15
4-HOPy∙∙∙SnF_4_	2.232	177.6	83.2	2.203	118.8	83.5	2.69
4-MePy∙∙∙SiF_4_	2.127	179.7	83.1	1.965	119.0	84.0	9.00
4-MePy∙∙∙GeF_4_	2.110	179.6	84.5	2.017	118.4	84.9	5.55
4-MePy∙∙∙SnF_4_	2.237	178.1	83.3	2.210	118.7	83.9	3.02
4-TFMPy∙∙∙SiF_4_	2.189	179.7	81.9	-	-	-	-
4-TFMPy∙∙∙GeF_4_	2.142	179.5	83.7	2.039	118.5	83.9	6.63
4-TFMPy∙∙∙SnF_4_	2.260	178.3	82.6	2.233	118.5	82.6	3.70

^a^ Energetic preference of axial over equatorial complex (kcal/mol).

**Table 3 molecules-24-00376-t003:** Difference in energy (kcal/mol) between axial and equatorial vacancy on an idealized trigonal bipyramid structure of TF_4_ molecule, along with maxima of molecular electrostatic potentials (kcal/mol) on the 0.001 a.u. isodensity surface.

Monomer	E(eq) − E(ax)	V_s,max_, Axial	V_s,max_, Equatorial
SiF_4_	22.56	120.6	126.5
GeF_4_	17.39	116.4	120.1
SnF_4_	10.68	124.3	131.4

**Table 4 molecules-24-00376-t004:** Interaction energies (E_int_, kcal mol^−1^) corrected for BSSE calculated at the MP2/cc-pVTZ (I), BLYP-D3/Def2TZVPP (II), and CCSD(T)/cc-pVTZ (III) levels of theory.

	Axial Complexes			Equatorial Complexes		
	(I)	(II)	(III)	(I)	(II)	(III)
Py∙∙∙SiF_4_	−26.75	−22.50	−27.07	−50.09	−47.38	−50.98
Py∙∙∙GeF_4_	−34.73	−34.37	−34.93	−52.02	−48.71	−52.62
Py∙∙∙SnF_4_	−39.68	−37.87	−39.85	−50.66	−44.97	−51.13
4-HOPy∙∙∙SiF_4_	−28.79	−22.50	−29.35	−53.52	−51.29	−54.78
4-HOPy∙∙∙GeF_4_	−36.62	−34.37	−37.10	−55.30	−51.93	−56.26
4-HOPy∙∙∙SnF_4_	−41.54	−37.87	−41.98	−52.86	−48.67	−53.63
4-MePy∙∙∙SiF_4_	−28.35	−25.97	−28.84	−52.10	−49.96	−53.25
4-MePy∙∙∙GeF_4_	−36.19	−36.29	−36.58	−53.98	−51.00	−54.84
4-MePy∙∙∙SnF_4_	−41.07	−39.70	−41.44	−52.22	−47.69	−52.93
4-TFMPy∙∙∙SiF_4_	−21.94	−9.43	−22.08	-	-	-
4-TFMPy∙∙∙GeF_4_	−30.62	−29.93	−30.67	−47.32	−42.98	−47.72
4-TFMPy∙∙∙SnF_4_	−35.81	−33.78	−35.83	−46.34	−41.53	−46.65

**Table 5 molecules-24-00376-t005:** Binding energies (ΔE_b_, kcal mol^−1^) calculated at the MP2/cc-pVTZ (I) and BLYP-D3/Def2TZVPP (II) levels of theory.

	Axial Complexes		Equatorial Complexes	
	(I)	(II)	(I)	(II)
Py∙∙∙SiF_4_	−10.39	−7.85	−0.98	1.67
Py∙∙∙GeF_4_	−20.51	−18.06	−14.60	−12.97
Py∙∙∙SnF_4_	−32.41	−27.64	−29.14	−24.59
4-HOPy∙∙∙SiF_4_	−11.13	−8.66	−2.53	−0.06
4-HOPy∙∙∙GeF_4_	−21.52	−19.29	−16.36	−14.86
4-HOPy∙∙∙SnF_4_	−33.65	−29.09	−30.96	−26.45
4-MePy∙∙∙SiF_4_	−11.06	−8.59	−2.06	0.31
4-MePy∙∙∙GeF_4_	−21.37	−19.15	−15.81	−14.44
4-MePy∙∙∙SnF_4_	−33.41	−28.89	−30.39	−26.20
4-TFMPy∙∙∙SiF_4_	−8.38	−6.36	-	-
4-TFMPy∙∙∙GeF_4_	−17.98	−15.41	−11.35	−9.67
4-TFMPy∙∙∙SnF_4_	−29.50	−24.66	−25.85	−21.16

**Table 6 molecules-24-00376-t006:** Deformation energies of complexes and maximum of MEP for TF_4_ in its geometry within the complex, calculated at the MP2/cc-pVTZ (I) level of theory. All quantities in kcal/mol.

	Axial Complexes			
	E_def_ of TF_4_	E_def_ of LB	Sum	V_s,max_
Py∙∙∙SiF_4_	20.35	0.35	20.70	93.4
Py∙∙∙GeF_4_	19.49	0.58	20.07	100.3
Py∙∙∙SnF_4_	12.64	0.74	13.38	111.3
4-HOPy∙∙∙SiF_4_	21.72	0.41	22.13	95.0
4-HOPy∙∙∙GeF_4_	20.43	0.67	21.10	101.0
4-HOPy∙∙∙SnF_4_	13.28	0.85	14.13	112.1
4-MePy∙∙∙SiF_4_	21.35	0.37	21.72	94.6
4-MePy∙∙∙GeF_4_	20.15	0.60	20.75	101.0
4-MePy∙∙∙SnF_4_	13.09	0.76	13.85	111.9
4-TFMPy∙∙∙SiF_4_	17.35	0.33	17.68	89.6
4-TFMPy∙∙∙GeF_4_	17.70	0.58	18.28	98.5
4-TFMPy∙∙∙SnF_4_	11.49	0.75	12.24	110.0
	**Equatorial Complexes**			
Py∙∙∙SiF_4_	53.05	0.82	53.87	107.5
Py∙∙∙GeF_4_	43.08	1.01	44.09	107.7
Py∙∙∙SnF_4_	26.69	1.00	27.69	119.5
4-HOPy∙∙∙SiF_4_	54.93	0.94	55.87	109.1
4-HOPy∙∙∙GeF_4_	44.65	1.15	45.80	108.9
4-HOPy∙∙∙SnF_4_	27.06	1.17	28.23	119.8
4-MePy∙∙∙SiF_4_	54.02	0.84	54.86	108.3
4-MePy∙∙∙GeF_4_	43.90	1.03	44.93	108.4
4-MePy∙∙∙SnF_4_	27.05	1.04	28.08	119.8
4-TFMPy∙∙∙SiF_4_	-	-	-	-
4-TFMPy∙∙∙GeF_4_	41.43	1.04	42.47	106.4
4-TFMPy∙∙∙SnF_4_	25.52	1.04	26.56	118.2

**Table 7 molecules-24-00376-t007:** EDA/BLYP-D3/ZORA/TZ2P decomposition of the interaction energy of complexes into Pauli repulsion (E_Pauli_), electrostatic (E_elec_), orbital interaction (E_oi_) and dispersion (E_disp_) components. All energies in kcal/mol.

	E_Pauli_			E_elec_			E_oi_			E_disp_		
	ax	eq	eq − ax	ax	eq	eq − ax	ax	eq	eq − ax	ax	eq	eq − ax
Py∙∙∙SiF_4_	89.72	117.10	27.38	−70.23	−93.70	−23.47	−40.00	−67.04	−27.04	−4.53	−4.38	0.15
Py∙∙∙GeF_4_	108.13	110.56	2.43	−84.50	−90.48	−5.98	−49.68	−64.82	−15.14	−4.63	−4.19	0.44
Py∙∙∙SnF_4_	99.74	87.36	−12.38	−85.48	−80.05	5.43	−45.43	−48.91	−3.48	−4.52	−3.98	0.54
4-HOPy∙∙∙SiF_4_	94.13	120.82	26.69	−74.10	−97.86	−23.76	−42.67	−70.50	−27.83	−4.53	−4.33	0.20
4-HOPy∙∙∙GeF_4_	112.02	116.15	4.13	−88.08	−95.58	−7.50	−51.93	−68.40	−16.47	−4.63	−4.20	0.43
4-HOPy∙∙∙SnF_4_	102.86	92.22	−10.64	−88.83	−85.27	3.56	−47.15	−52.27	−5.12	−4.52	−3.93	0.59
4-MePy∙∙∙SiF_4_	92.92	117.10	24.18	−73.01	−93.70	−20.60	−42.06	−67.04	−24.98	−4.54	−4.38	0.16
4-MePy∙∙∙GeF_4_	110.73	110.56	−0.17	−86.99	−90.48	−3.49	−51.39	−64.82	−13.43	−4.65	−4.19	0.46
4-MePy∙∙∙SnF_4_	101.57	87.36	−14.21	−87.63	−80.05	7.58	−46.71	−48.91	−2.20	−4.54	−3.98	0.56
4-TFMPy∙∙∙SiF_4_	80.38	-	-	−61.74	-	-	−34.59	-	-	−4.56	-	-
4-TFMPy∙∙∙GeF_4_	101.43	104.42	2.99	−77.67	−83.12	−5.45	−45.96	−60.09	−14.13	−4.66	−4.26	0.40
4-TFMPy∙∙∙SnF_4_	94.96	82.47	−12.49	−79.49	−73.92	5.57	−42.96	−46.70	−3.74	−4.56	−3.93	0.63

**Table 8 molecules-24-00376-t008:** AIM bond critical point (BCP) properties: electron density *ρ*, Laplacian of electron density ∇^2^*ρ* (both in atomic units) and total electron energy (H, a.u.) obtained at the MP2/cc-pVTZ level.

	Axial Complexes				Equatorial Complexes			
	Interaction	*ρ*	∇^2^*ρ*	H	Interaction	*ρ*	∇^2^*ρ*	H
Py∙∙∙SiF_4_	Si···N	0.054	0.145	−0.022	Si···N	0.077	0.276	−0.032
	F···H	0.015	0.073	0.003	F···H	0.019	0.094	0.003
	-	-	-		F···H	0.019	0.094	
Py∙∙∙GeF_4_	Ge···N	0.077	0.166	−0.030	Ge···N	0.094	0.216	−0.042
	F···H	0.015	0.068	0.003	F···H	0.020	0.093	0.004
	F···H	0.015	0.070	0.003	F···H	0.020	0.093	0.004
Py∙∙∙SnF_4_	Sn···N	0.073	0.211	−0.020	Sn···N	0.077	0.230	−0.022
	F···H	0.014	0.059	0.002	F···H	0.021	0.095	0.004
	F···H	0.014	0.059	0.002	F···H	0.021	0.095	0.004
4-HOPy∙∙∙SiF_4_	Si···N	0.057	0.159	−0.023	Si···N	0.079	0.289	−0.033
	F···H	0.015	0.074	0.003	F···H	0.021	0.100	0.004
	-	-	-		F···H	0.020	0.099	
4-HOPy∙∙∙GeF_4_	Ge···N	0.079	0.174	−0.032	Ge···N	0.097	0.225	−0.044
	F···H	0.015	0.070	0.003	F···H	0.021	0.100	0.004
	F···H	0.015	0.068	0.003	F···H	0.021	0.099	0.004
4-HOPy∙∙∙SnF_4_	Sn···N	0.075	0.218	−0.021	Sn···N	0.079	0.240	−0.023
	F···H	0.014	0.059	0.002	F···H	0.021	0.095	0.004
	F···H	0.014	0.060	0.002	F···H	0.020	0.093	0.003
4-MePy∙∙∙SiF_4_	Si···N	0.056	0.155	−0.022	Si···N	0.079	0.284	−0.032
	F···H	0.015	0.073	0.003	F···H	0.019	0.095	0.003
	-	-	-		F···H	0.019	0.094	
4-MePy∙∙∙GeF_4_	Ge···N	0.079	0.170	−0.032	Ge···N	0.096	0.221	−0.043
	F···H	0.015	0.069	0.003	F···H	0.020	0.094	0.004
	F···H	0.015	0.069	0.003	F···H	0.020	0.094	0.004
4-MePy∙∙∙SnF_4_	Sn···N	0.075	0.215	−0.021	Sn···N	0.078	0.235	−0.022
	F···H	0.013	0.058	0.002	F···H	0.020	0.093	0.003
	F···H	0.013	0.058	0.002	F···H	0.020	0.093	0.003
4-TFMPy∙∙∙SiF_4_	Si···N	0.049	0.118	−0.020	-	-	-	
4-TFMPy∙∙∙GeF_4_	Ge···N	0.073	0.158	−0.027	Ge···N	0.091	0.208	−0.040
	F···H	0.015	0.068	0.003	F···H	0.020	0.094	0.003
	F···H	0.015	0.068	0.003	F···H	0.020	0.094	0.003
4-TFMPy∙∙∙SnF_4_	Sn···N	0.071	0.204	−0.019	Sn···N	0.075	0.222	−0.020
	F···H	0.014	0.060	0.002	F···H	0.021	0.096	0.004
	F···H	0.014	0.059	0.002	F···H	0.021	0.096	0.004

**Table 9 molecules-24-00376-t009:** NBO values of sum of E(2) (kcal/mol) for LP (N) → LP* (T), and total charge transfer (CT, me) from LB to TF_4_ obtained at the BLYP-D3/def2-TVZPP level.

	Axial Complexes		Equatorial Complexes	
	∑ LP (N) → LP* (T)	CT	∑ LP (N) → LP* (T)	CT
Py∙∙∙SiF_4_	72.8	121	161.2	212
Py∙∙∙GeF_4_	137.7	174	158.4	220
Py∙∙∙SnF_4_	108.2	185	111.0	202
4-HOPy∙∙∙SiF_4_	98.5	139	169.0	221
4-HOPy∙∙∙GeF_4_	146.0	182	166.2	227
4-HOPy∙∙∙SnF_4_	113.5	191	116.6	210
4-MePy∙∙∙SiF_4_	90.7	138	166.9	218
4-MePy∙∙∙GeF_4_	143.8	180	163.6	225
4-MePy∙∙∙SnF_4_	110.4	190	114.9	209
4-TFMPy∙∙∙SiF_4_	17.9	36	-	-
4-TFMPy∙∙∙GeF_4_	123.4	159	149.2	206
4-TFMPy∙∙∙SnF_4_	100.6	172	103.5	192

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
