# Peer review of "Dual Geometry Schemes in Tetrel Bonds: Complexes between TF4 (T = Si, Ge, Sn) and Pyridine Derivatives"

_molecules, 2019, doi:10.3390/molecules24020376_

Reviewer 1 Report

This paper describes carefully deformation of tetrahedral molecules such as TF4 (T=Si, Ge, Sn), 14 group atoms, with an N-base. Computational scheme in the present study is adequate, i.e., geometry optimization using MP2/cc-pVTZ, BLAYP-D3/Def2TZVPP, and CCSD(T)/cc-pVTZ level of theory with BSSE correction. Energy decomposition analysis, natural bond orbital analysis, and electron density topology analysis are performed to realize bonding nature. The present results and their interpretation may be good. One thing is that this paper become much better if a series of results for CF4 is present in the main text. Because a carbon atom is the lightest for the 14 group element and more familiar with the target molecules in the present manuscript.

Except for the above thing, I recommend this paper is accepted to Molecules.

Minor points:

Line 9, page 1: “a N-base” should be “an N-base”.

Line 23, page 1: “An important element …” this sentence is not proper.

Author Response

This paper describes carefully deformation of tetrahedral molecules such as TF4 (T=Si, Ge, Sn), 14 group atoms, with an N-base. Computational scheme in the present study is adequate, i.e., geometry optimization using MP2/cc-pVTZ, BLAYP-D3/Def2TZVPP, and CCSD(T)/cc-pVTZ level of theory with BSSE correction. Energy decomposition analysis, natural bond orbital analysis, and electron density topology analysis are performed to realize bonding nature. The present results and their interpretation may be good. One thing is that this paper become much better if a series of results for CF4 is present in the main text. Because a carbon atom is the lightest for the 14 group element and more familiar with the target molecules in the present manuscript.

A paragraph has been added at the end of the Discussion section in which the reluctance of CF4 to engage in tetrel bonding is discussed.

Except for the above thing, I recommend this paper is accepted to Molecules.

Minor points:

Line 9, page 1: “a N-base” should be “an N-base”.

Line 23, page 1: “An important element …” this sentence is not proper.

We would like to thank Reviewer for pointing this out. The appropriate corrections have been made in text.

Reviewer 2 Report

Wiktor Zierkiewicz and others presented a nice  study with the title "Dual Geometry Schemes in Tetrel Bonds: Complexes  between TF4  (T = Si, Ge, Sn) and Pyridine Derivatives". The paper is about the tetrel bond and it focuses on an original aspect, which is the possible presence of two different isomers of the TF4-pyridine adduct.

The work is well thought, written and presented, thanks to a very detailed and exhaustive computational study. The authors analyze the system from different pont of view (NBO, AIM, EDA...), gaining many information.

I would say the paper is suitable for publication in Molecules, but a point should be clarified.

The authors refer to tetrel bond as a "noncovalent" interaction, but their results seem to tell a different story: from EDA, the orbital contribution to the interaction is between -42 and -70 kcal/mol for the equatorial and axial geometries (and, interestigly, lower for eq-ax geometries), from NBO, the charge transfer of 0.2 electrons and the E(2) energy is quite high... I would say the tetrel bond has a non-negligible covalent nature!

In the Discussion section, this aspect should be faced, also comparing the infomation obtained with different methods. I underline that reference n.3 is titled "What is the covalency of hydrogen bonding?", indication that the problem exists for weak interactions and it is not completely solved... especially for halogen/chalcogen/pnicogen/tetrel bonds!

Table 7 is almost unreadable, but I trust in the Editorial Office for a better layout

Author Response

Wiktor Zierkiewicz and others presented a nice  study with the title "Dual Geometry Schemes in Tetrel Bonds: Complexes  between TF4  (T = Si, Ge, Sn) and Pyridine Derivatives". The paper is about the tetrel bond and it focuses on an original aspect, which is the possible presence of two different isomers of the TF4-pyridine adduct.

The work is well thought, written and presented, thanks to a very detailed and exhaustive computational study. The authors analyze the system from different pont of view (NBO, AIM, EDA...), gaining many information.

I would say the paper is suitable for publication in Molecules, but a point should be clarified.

The authors refer to tetrel bond as a "noncovalent" interaction, but their results seem to tell a different story: from EDA, the orbital contribution to the interaction is between -42 and -70 kcal/mol for the equatorial and axial geometries (and, interestigly, lower for eq-ax geometries), from NBO, the charge transfer of 0.2 electrons and the E(2) energy is quite high... I would say the tetrel bond has a non-negligible covalent nature! 

In the Discussion section, this aspect should be faced, also comparing the infomation obtained with different methods. I underline that reference n.3 is titled "What is the covalency of hydrogen bonding?", indication that the problem exists for weak interactions and it is not completely solved... especially for halogen/chalcogen/pnicogen/tetrel bonds!

We agree that in the case of strong noncovalent interactions such as these the partial covalent nature is relevant and worth discussion.  A summarizing paragraph to this effect has therefore been added at the conclusion of the Analysis section.

Table 7 is almost unreadable, but I trust in the Editorial Office for a better layout

To make it more readable Table 7 has been re-edited (smaller font size).

Reviewer 3 Report

The manuscript written by Zierkiewicz, Scheiner and coworkers deals with geometry schemes in tetrel bonds. They used a high level of accuracy to perform their calculations (MP2) together with large basis set. This work was undoubtedly realized with great care and the manuscript is very clear to this referee. Therefore, the article can be published in the present form. However, according to this referee, the story can be extended and improved by exploring the following suggestions:

Authors should give a chance to the B2PLYP functional (double hybrid functional containing a non-negligible amount of MP2)

Simulate the vibrational (an-)harmonic spectra of all the compounds (cf. Barone et al : JCTC 2014, 10, 5586; JCTC 2015, 11, 4364; JCTC 2010, 6, 2115)

Provide the imaginary frequencies of transition states (Fig 3)

Adjust written part into the Figures to keep a more pleasant picture

Author Response

The manuscript written by Zierkiewicz, Scheiner and coworkers deals with geometry schemes in tetrel bonds. They used a high level of accuracy to perform their calculations (MP2) together with large basis set. This work was undoubtedly realized with great care and the manuscript is very clear to this referee. Therefore, the article can be published in the present form. However, according to this referee, the story can be extended and improved by exploring the following suggestions:

Authors should give a chance to the B2PLYP functional (double hybrid functional containing a non-negligible amount of MP2)

  According to this reviewer’s request that we test out another DFT functional, we recalculated the interaction energies for several of our dimers.  As discussed in the penultimate paragraph of the Discussion section, and a new Table S4, we found this functional to have a certain amount of value, but its computational cost would not justify its use on a routine basis.

Simulate the vibrational (an-)harmonic spectra of all the compounds (cf. Barone et al : JCTC 2014, 10, 5586; JCTC 2015, 11, 4364; JCTC 2010, 6, 2115)

We had indeed performed calculations of the vibrational spectra of all the compounds investigated, as described in the Methods section, in order to verify the character of our structures as true minima. A detailed presentation concerning all of the many IR spectral lines is not a focus of this work, and would divert a reader from our central message.  And indeed, calculation of anharmonic corrections is fraught with complications, and the appropriate method to do so is a controversial topic.  As such, an analysis of this type would represent a topic for one or even more separate papers, and is clearly beyond the scope of the present work.

 Provide the imaginary frequencies of transition states (Fig 3)

A table containing these imaginary frequencies of transition states has been added to the SI (Table S2), with an appropriate reference to it in the revised text.

Adjust written part into the Figures to keep a more pleasant picture

This request is very vague, and we are unsure just what is being requested here.  Which figure and what specific changes are needed?  After reevaluating all of the figures, we are confident they are already quite clear.

Reviewer 4 Report

I think that this manuscript can be accepted for publucation in Molecules.

The work is devoted to quite relevant, modern and fashionable topic (non-covalent interactions).

The level of theory used is quite good, various approaches and methods of computational studies were applied (calculations of molecular electrostatic potential and interaction energies; EDA, NBO, and AIM analyses). The obtained results are presented quite clearly.

The manuscript may be of interest to specialists in the field of weak supramolecular interactions.

Suggestion for authors:
The Cartesian atomic coordinates for optimized equilibrium geometries of all model species should be given in SUPPLEMENTARY INFORMATION.

Author Response

I think that this manuscript can be accepted for publucation in Molecules.

The work is devoted to quite relevant, modern and fashionable topic (non-covalent interactions).

The level of theory used is quite good, various approaches and methods of computational studies were applied (calculations of molecular electrostatic potential and interaction energies; EDA, NBO, and AIM analyses). The obtained results are presented quite clearly.

The manuscript may be of interest to specialists in the field of weak supramolecular interactions.

Suggestion for authors:
The Cartesian atomic coordinates for optimized equilibrium geometries of all model species should be given in SUPPLEMENTARY INFORMATION.

According to the Reviewer suggestion the coordinates have been added in SI (Table S5).

Reviewer 5 Report

The current paper represents an interesting theoretical study of complexes between TF4 (T = Si, Ge, Sn) and pyridine derivatives. The authors mainly focused on the theory of chemical bonding in these systems using the different computational approaches like DFT, MP2, AIM, NBO etc. Such comprehensive analysis provides a real understanding how these complexes are formed and how the properties of the complexes depend on the type of T-atom. I just want to recommend authors some points that are important for the improvement of MS:

1) Fist and most desirable point relates to the degree of covalency for the studied T-N bonds. I recommend to analyze the delocalization indexes (DI) values taken from AIM analysis. These indexes quantitatively demonstrate the number of electrons delocalized between the interacting atoms. More information about DI can be found in the original Bader's monograph as well as in some recent publications (DOI: 10.1134/S1070363212070122, 10.1039/C4RA13806F)  that should be introduced into the list of references.

2) I.m surprised why authors do note refer t the Bader's classical papers like:

R.F.W. Bader, Atoms Inmolecules. A Quantum Theory, Clarendon Press, Oxford, 1990
R.F.W. Bader, J. Phys. Chem. A 102 (1998) 7314.
R.F.W. Bader, H. Essen, J. Chern. Phys. 80 (1984) 1943.
R.F.W. Bader, T.S. Slee, D. Cremer, E. Kraka, J. Am. Chem. Soc. 105 (1983) 5061.

Please, add these refs into the paragraph 3.4 when analyze electron density peculiarities in the corresponding critical points (CPs).

3) What is the "total electron energy (H, kcal/mol)"? Is it a sum of potential (V) and kinetic (G) energy densities? if yes, the authors should mention the original paper by Cremer and Kraka (Croat. Chem. Acta. 57 (1984) 1259) who developed this term for the clear classification of chemical bonds depending on the V vs. H balance. Moreover, H values (as well as V and G) are usually presented in the literature in a.u. (see in refs. before).

4) Paragraph 3.5. What is the relative Gibbs free energy for the TS an Eq structures relative to the Ax structure. Did authors perform the TS founding? If yes, what is the method was used?

Finally, I recommend the minor revision of MS with the subsequent revision of improved MS.

Author Response

The current paper represents an interesting theoretical study of complexes between TF4 (T = Si, Ge, Sn) and pyridine derivatives. The authors mainly focused on the theory of chemical bonding in these systems using the different computational approaches like DFT, MP2, AIM, NBO etc. Such comprehensive analysis provides a real understanding how these complexes are formed and how the properties of the complexes depend on the type of T-atom. I just want to recommend authors some points that are important for the improvement of MS:

1) Fist and most desirable point relates to the degree of covalency for the studied T-N bonds. I recommend to analyze the delocalization indexes (DI) values taken from AIM analysis. These indexes quantitatively demonstrate the number of electrons delocalized between the interacting atoms. More information about DI can be found in the original Bader's monograph as well as in some recent publications (DOI: 10.1134/S1070363212070122, 10.1039/C4RA13806F)  that should be introduced into the list of references.

We are thankful for this reviewer’s suggestion about DI. We present the results of this parameter in a separate table in the SI (Table S1), along with an evaluation of its meaning at the end of the discussion of AIM data.

2) Im surprised why authors do note refer t the Bader's classical papers like:

R.F.W. Bader, Atoms Inmolecules. A Quantum Theory, Clarendon Press, Oxford, 1990
R.F.W. Bader, J. Phys. Chem. A 102 (1998) 7314.
R.F.W. Bader, H. Essen, J. Chern. Phys. 80 (1984) 1943.
R.F.W. Bader, T.S. Slee, D. Cremer, E. Kraka, J. Am. Chem. Soc. 105 (1983) 5061.

Please, add these refs into the paragraph 3.4 when analyze electron density peculiarities in the corresponding critical points (CPs).

These references have been added. The citations are at the beginning of Section 2.4.

3) What is the "total electron energy (H, kcal/mol)"? Is it a sum of potential (V) and kinetic (G) energy densities? if yes, the authors should mention the original paper by Cremer and Kraka (Croat. Chem. Acta. 57 (1984) 1259) who developed this term for the clear classification of chemical bonds depending on the V vs. H balance. Moreover, H values (as well as V and G) are usually presented in the literature in a.u. (see in refs. before).

Yes, the Reviewer is correct, H = V + G. The reference mentioned by this reviewer has been added at the beginning of our AIM discussion.  All H values units have been converted to atomic units for purposes of consistency.

4) Paragraph 3.5. What is the relative Gibbs free energy for the TS an Eq structures relative to the Ax structure. Did authors perform the TS founding? If yes, what is the method was used?

Calculated Gibbs free energies of axial and equatorial conformers as well as TS have been collected in table S2. Details about TS founding have been added in the Methods section.

Finally, I recommend the minor revision of MS with the subsequent revision of improved MS.